# Topology of Reasoning: Retrieved Cell Complex-Augmented Generation for Textual Graph Question Answering

**Sen Zhao**[*][†][1]**, Lincheng Zhou**[*][2]**, Yue Chen**[2]**, and Ding Zou**[3]

[1] Academy of Advanced Interdisciplinary Studies, Chongqing University of Posts and Telecommunications Chongqing, China
[2] School of Computer Science and Technology, Chongqing University of Posts and Telecommunications Chongqing, China
[3] Intelligent System Department, Zhongxing Telecom Equipment (ZTE), Changsha, Hunan, China

## Abstract

Retrieval-Augmented Generation (RAG) enhances the reasoning ability of Large Language Models (LLMs) by dynamically integrating external knowledge, thereby mitigating hallucinations and strengthening contextual grounding for structured data such as graphs. Nevertheless, most existing RAG variants for textual graphs concentrate on low-dimensional structures—treating nodes as entities (0-dimensional) and edges or paths as pairwise or sequential relations (1-dimensional), but overlook cycles, which are crucial for reasoning over relational loops. Such cycles often arise in questions requiring closed-loop inference about similar objects or relative positions. This limitation often results in incomplete contextual grounding and restricted reasoning capability. In this work, we propose **Topo**logy-enhanced **R**etrieval-**A**ugmented **G**eneration (TopoRAG), a novel framework for textual graph question answering that effectively captures higher-dimensional topological and relational dependencies. Specifically, TopoRAG first lifts textual graphs into cellular complexes to model multi-dimensional topological structures. Leveraging these lifted representations, a topology-aware sub-complex retrieval mechanism is proposed to extract cellular complexes relevant to the input query, providing compact and informative topological context. Finally, a multi-dimensional topological reasoning mechanism operates over these complexes to propagate relational information and guide LLMs in performing structured, logic-aware inference. Empirical evaluations demonstrate that our method consistently surpasses existing baselines across diverse textual graph tasks.

## 1 Introduction

Large Language Models (LLMs) exhibit strong language understanding and generation capabilities, but their reliance on pre-training corpora—limited in scope and timeliness—often leads to hallucinations, producing inaccurate or fabricated content that challenges knowledge-intensive reasoning Huang et al. (2023b). To mitigate these issues, Retrieval-Augmented Generation (RAG) has recently emerged as an effective approach Fan et al. (2024); Sun et al. (2024); Baek et al. (2023); Sen et al. (2023), dynamically retrieving relevant external knowledge and incorporating it into the generation process. By enhancing contextual grounding and factual accuracy, RAG improves reasoning over structured data and reduces hallucination Gao et al. (2023). However, traditional RAG methods often overlook the structured dependencies among textual entities and struggle to capture global relational patterns, limiting their applicability for graph-structured reasoning tasks.

To address these challenges, Graph Retrieval-Augmented Generation (GraphRAG) Edge et al. (2024); Hu et al. (2024); Mavromatis & Karypis (2025) extends conventional RAG by retrieving

---

[*]Contributing equally.
[†]Correspondence to Sen Zhao <zhaosen@cqupt.edu.cn>.

not only documents but also graph elements, which provide richer relational context for reasoning over textual graphs. *G-Retriever* He et al. (2024) introduces the first general GraphRAG framework for textual graphs, formulating retrieval as a Prize-Collecting Steiner Tree problem to extract compact and relevant subgraphs. GNN-RAG Mavromatis & Karypis (2025) and SubgraphRAG Li et al. (2025) further develop specialized retrieval modules to extract subgraphs from knowledge graphs. However, existing approaches, primarily operate on low-dimensional elements and largely ignore higher-dimensional topological structures such as cycles, which are crucial for reasoning over relational loops and complex dependencies in textual graphs. This limitation becomes more evident when considering the structural nature of reasoning itself. Recent empirical studies (Minegishi et al. (2025)) analyzing reasoning processes in large language models observe that stronger reasoning behaviors often involve recurrent dependency patterns that resemble cyclic structures in reasoning graphs. Such observations suggest that effective reasoning may inherently rely on non-linear relational organization, motivating the need to explicitly model cyclic dependencies in textual graphs.

In many real-world textual graphs, essential information arises not only from nodes (0-cells) that encode entity attributes, edges and paths (1-cells) representing pairwise or multi-hop relations, but also from cycles (2-cells) capturing higher-dimensional dependencies. As illustrated in Fig. 1, reasoning over 0-cells enables answering simple attribute-based questions (Fig. 1 a), while incorporating 1-cells supports inference over one-hop to multi-hop relational queries (Fig. 1 b). However, certain queries require cyclic dependencies

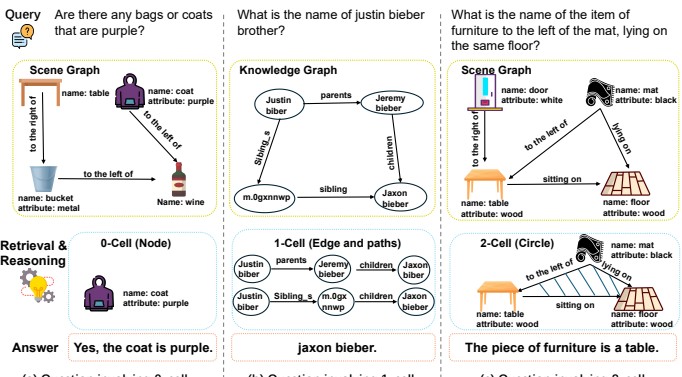

Figure 1: Illustration of question answering with varying dimensional topological characteristics.

among multiple entities, where the answer emerges only from reasoning over 2-cells. For example, the question in Fig. 1 (c) involves a closed relational loop that links spatial relations with material consistency, which cannot be resolved by nodes and edges alone. Capturing structural information across multiple topological dimensions provides indispensable context for structured logical inference, as higher-dimensional dependencies complement lower-dimensional relations to enable reasoning beyond simple pairwise interactions. Consequently, retrieval and reasoning mechanisms that explicitly incorporate multi-dimensional topological features are essential for understanding and answering questions over complex textual graphs.

In this work, we propose **Topo**logy-enhanced **R**etrieval-**A**ugmented **G**eneration (**TopoRAG**), a novel framework for textual graph question answering that explicitly models higher-dimensional topological and relational dependencies. Specifically, TopoRAG first lifts input textual graphs into cellular complexes to capture multi-dimensional topological structures, including cycles that encode closed-loop dependencies critical for relational reasoning. Leveraging these lifted representations, a topology-aware subcomplex retrieval mechanism is introduced to extract cellular complexes that are most relevant to the input query, providing compact yet informative topological context for downstream reasoning. Furthermore, a multi-dimensional topological reasoning mechanism operates over the retrieved complexes to propagate relational information across different topological dimensions, enabling structured, logic-aware inference that naturally integrates with LLM reasoning. Extensive experiments demonstrate that TopoRAG consistently outperforms state-of-the-art baselines.

## 2 RELATED WORKS

Large Language Models (LLMs) have shown impressive capabilities in language understanding and text generation, yet they remain constrained by the boundaries of their pre-training corpus, lacking domain-specific expertise, real-time updates, and proprietary knowledge. These limitations

frequently manifest as hallucinations, where models produce inaccurate or fabricated content Huang et al. (2023b). To address this issue, Retrieval-Augmented Generation (RAG) Fan et al. (2024); Sun et al. (2024); Baek et al. (2023); Sen et al. (2023) has emerged as a promising paradigm.

RAG enhances LLMs by dynamically retrieving relevant external knowledge and incorporating it into the generation process, thereby improving factual accuracy, contextual grounding, and interpretability Gao et al. (2023). Nevertheless, existing RAG methods are not without shortcomings in real-world applications. They often overlook structured dependencies among textual entities, rely on lengthy concatenated snippets that may obscure critical information (the "lost in the middle" problem Liu et al. (2024)), and struggle to capture global structural patterns essential for tasks such as query-focused summarization.

To address these challenges, Graph Retrieval-Augmented Generation (GraphRAG) Edge et al. (2024); Hu et al. (2024); Mavromatis & Karypis (2025) extends conventional RAG by retrieving not only documents but also graph elements such as nodes, triples, and subgraphs. Building on this idea, G-Retriever He et al. (2024) introduces the first general RAG framework for textual graphs, formulating retrieval as a Prize-Collecting Steiner Tree problem to extract compact and relevant subgraphs. GNN-RAG Mavromatis & Karypis (2025) improves knowledge graph QA by integrating GNN-based representations for task-specific subgraph selection, and SubgraphRAG Li et al. (2025) incorporates lightweight triple scoring and distance encoding to achieve efficient subgraph retrieval. Nevertheless, existing methods overlook high-dimensional cyclic dependencies, motivating our approach to incorporate multi-dimensional cell structures for enhanced retrieval and reasoning.

We also discuss related works on graphs & LLMs, and topological deep learning in Appendix B.

## 3 PRELIMINARIES

***Definition 1. (Cell Complex Hansen & Ghrist (2019)).*** A **regular cell complex** is a topological space $X$ decomposed into a collection of disjoint subspaces $\{x_\sigma\}_{\sigma \in P_X}$, referred to as *cells*, satisfying the following conditions:

1. For each point $p \in X$, $\exists$ an open neighborhood intersects only finitely many cells.
2. For any pair of cells $x_\sigma, x_\tau$, the intersection $x_\tau \cap \overline{x_\sigma}$ is nonempty if and only if $x_\tau \subseteq \overline{x_\sigma}$, where $\overline{x_\sigma}$ denotes the topological closure of $x_\sigma$.
3. Each cell $x_\sigma$ is homeomorphic to an open ball in $\mathbb{R}^n$ for some non-negative integer $n$.
4. *(Regularity)* The closure $\overline{x_\sigma}$ of every cell is homeomorphic to a closed ball in $\mathbb{R}^{n_\sigma}$, with the interior mapped homeomorphically onto $x_\sigma$ itself.

***Definition 2.*** A **cellular lifting map** is a function $f : \mathcal{G} \to X$ from the space of graphs $\mathcal{G}$ to the space of regular cell complexes $X$, satisfying that two graphs $G_1, G_2 \in \mathcal{G}$ are isomorphic if and only if their corresponding cell complexes $f(G_1)$ and $f(G_2)$ are isomorphic. Intuitively, a cell complex is built hierarchically by first considering 0-cells (vertices), then attaching 1-cells (edges) via their endpoints, and further incorporating higher-dimensional cells by gluing disks along cycles.

***Definition 3. (Retrieved Cell Complex-Augmented Question Answering).*** Given a textual graph $\mathcal{G} = (V, E, \{t_n\}_{n \in V}, \{t_e\}_{e \in E})$, where each node $n \in V$ and edge $e \in E$ is associated with textual attributes $t_n \in D^{L_n}$ and $t_e \in D^{L_e}$, we lift $\mathcal{G}$ into a regular cell complex $X$ through a cellular lifting map $f : \mathcal{G} \to X$. The resulting complex $X = \{x_\sigma\}$ contains multi-dimensional structures, including 0-cells (nodes), 1-cells (edges/paths), and higher-dimensional cells (e.g., 2-cells as cycles).

To enable retrieval, a query $Q$ is first encoded by a language model into a dense representation:

$$z_Q = \text{LM}(Q) \in \mathbb{R}^d. \tag{1}$$

Each cell $x_\sigma \in X$ is also represented by an embedding $z_\sigma$, obtained from its textual attributes together with a topological descriptor $z_\sigma^d$ that summarizes its $d$-dimensional structure. We then apply a top-$k$ similarity-based retrieval strategy to select the most relevant cells:

$$X_k = \text{TopK}_{x_\sigma \in X} \ \cos(z_Q, z_\sigma^d), \tag{2}$$

where $\cos(\cdot, \cdot)$ denotes cosine similarity. This step yields a candidate set of cells $X_k = \{x_{\sigma_1}, \ldots, x_{\sigma_k}\}$ spanning multiple topological dimensions.

The task is defined as follows: given a natural language query $Q$ and the lifted cell complex $X$, the model must retrieve the most relevant subcomplexes $X^*$ and reason over their multi-dimensional structures to generate an answer $A$. Formally, the QA function is

$$f : (\mathcal{G}, X, Q) \mapsto A, \tag{3}$$

where $A$ is a natural language sequence generated by the LLM under the conditional likelihood

$$p_\theta(A \mid [P_e; Q; X^*]) = \prod_{i=1}^{|A|} p_\theta(a_i \mid a_{<i}, [P_e; Q; X^*]). \tag{4}$$

Here, $[P_e; Q; X^*]$ denotes the concatenation of soft prompt embeddings $P_e$, query tokens, and retrieved subcomplex representations, while $a_{<i}$ represents the prefix of $A$ up to step $i-1$.

Training proceeds by maximizing the likelihood of the ground-truth answer $A^*$:

$$\max_{P_e} \log p_\theta(A^* \mid [P_e; Q; X^*]), \tag{5}$$

where only the soft prompt parameters $P_e$ are updated, while the LLM parameters $\theta$ remain fixed.

## 4 THE TOPORAG FRAMEWORK

In this section, we present the architecture of **TopoRAG** (illustrated in Figure 2), a topology-enhanced retrieval-augmented generation framework designed for textual graph question answering. TopoRAG is composed of four key components. First, the *Cellular Representation Lifting* module transforms input textual graphs into regular cell complexes, providing expressive multi-dimensional topological representations that go beyond nodes and edges. Second, the *Topology-aware Subcomplex Retrieval* module identifies the most relevant subcomplexes with respect to the query by jointly considering semantic similarity and topological structure. Third, the *Multi-dimensional Topological Reasoning* module propagates relational information across different topological dimensions, enabling structured and logic-aware inference. Finally, the *Cell Complex-Augmented Generation* module integrates retrieved subcomplex representations into the LLM to guide answer generation, ensuring faithful and topology-consistent responses.

### 4.1 CELLULAR REPRESENTATION LIFTING

Given a textual graph $\mathcal{G} = (V, E, \{t_n\}_{n \in V}, \{t_e\}_{e \in E})$, we aim to lift it into a higher-dimensional topological space that faithfully encodes both relational and structural dependencies. This is achieved by constructing a *regular cell complex* $X$ through a cellular lifting map $f : \mathcal{G} \to X$.

We first regard $\mathcal{G}$ as a 1-dimensional cell complex, where each vertex $v \in V$ corresponds to a 0-cell $x_v^0 \in X^{(0)}$, and each edge $(u, v) \in E$ corresponds to a 1-cell $x_{(u,v)}^1 \in X^{(1)}$ attached to its endpoint 0-cells $x_u^0$ and $x_v^0$. This forms the *cellular 1-skeleton*:

$$X^{(1)} = X^{(0)} \cup \{x_{(u,v)}^1 \mid (u, v) \in E\}. \tag{6}$$

To elaborate, consider $t_v \in D^{L_v}$ as the text attributes of vertex $v$ and $t_{(u,v)} \in D^{L(u,v)}$ as those of edge $(u, v)$. Utilizing a pre-trained LM, such as SentenceBert (Reimers & Gurevych, 2019), we apply the LM to these attributes, yielding the representations:

$$z_v^0 = \text{LM}(t_v) \in \mathbb{R}^d, \quad z_{(u,v)}^1 = \text{LM}(t_{(u,v)}) \in \mathbb{R}^d, \tag{7}$$

where $d$ denotes the dimension of the output vector. This yields cellular embeddings for both nodes and edges, which serve as the lifted representation in the subsequent module.

To incorporate high-dimensional topological structures, we extend $X^{(1)}$ by identifying fundamental cycles. Specifically, we fix a spanning tree $\mathcal{T} \subseteq G$ and apply the quotient map:

$$\gamma : G \to G/\mathcal{T}, \tag{8}$$

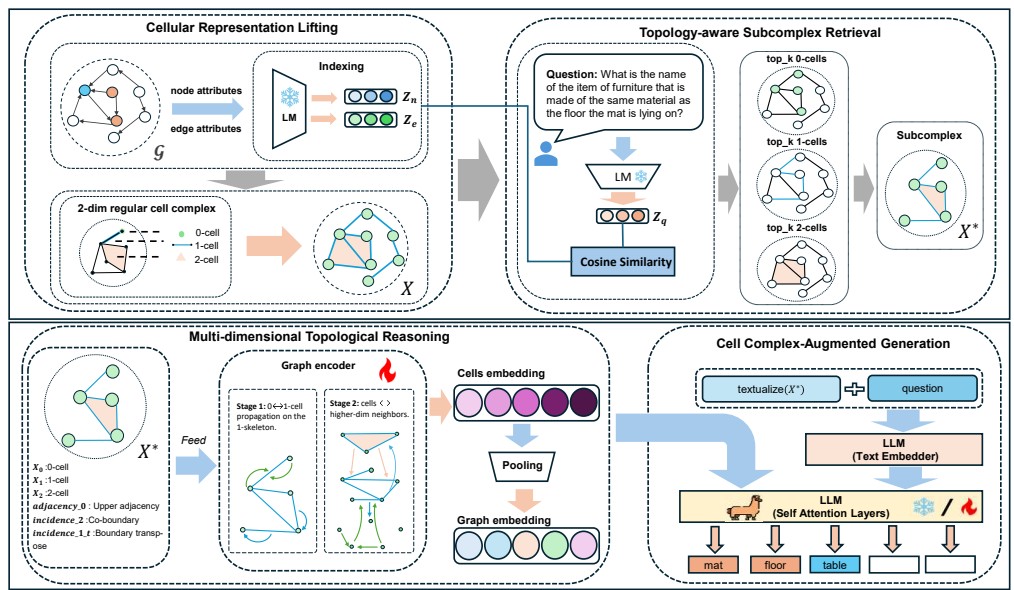

Figure 2: The overview of Topology-enhanced Retrieval-Augmented Framework.

which collapses $\mathcal{T}$ to a single point. Each non-tree edge $e = (u,v) \in E \setminus \mathcal{T}$ then induces a fundamental cycle by connecting $e$ with the unique path in $\mathcal{T}$ between $u$ and $v$. For every such cycle, we attach a 2-cell $x_e^2 \in X^{(2)}$ via the attaching map

$$\varphi_\alpha : \partial D^2 \cong S^1 \to G^{(1)}, \tag{9}$$

that glues the boundary of a disk $D^2$ to the cycle. Formally, the set of 2-cells is

$$X^{(2)} = \{x_e^2 \simeq D^2 \mid e \in E \setminus \mathcal{T}\}. \tag{10}$$

The resulting cell complex $X = X^{(0)} \cup X^{(1)} \cup X^{(2)}$ augments the original graph with multi-dimensional topological structures.

**Proposition 1.** *$G/\mathcal{T}$ is homotopy-equivalent to $G$, and $\gamma$ induces an isomorphism on the first homology group $H_1(G; \mathbb{Z})$.*

*Proof.* **Contractibility of the spanning tree.** A spanning tree $\mathcal{T}$ is connected and acyclic, hence it is contractible. In topological terms, a contractible subspace can be continuously shrunk to a point within itself.

**Collapsing a contractible subspace.** Consider the quotient map $\gamma : G \to G/\mathcal{T}$ that identifies all points of $\mathcal{T}$ to a single vertex $v_0$. Collapsing a contractible subspace is a deformation retraction up to homotopy: there exists a continuous map $r : G \to G/\mathcal{T}$ and a homotopy $H : G \times [0,1] \to G$ such that $H(x,0) = x$ and $H(x,1) = r(x)$ for all $x \in G$, with $r \circ \gamma \simeq \mathrm{id}_{G/\mathcal{T}}$. Therefore, $G$ and $G/\mathcal{T}$ are homotopy-equivalent.

**Induced isomorphism on first homology.** Homotopy-equivalent spaces have isomorphic homology groups. Hence the induced map $\gamma_* : H_1(G; \mathbb{Z}) \to H_1(G/\mathcal{T}; \mathbb{Z})$ is an isomorphism.

**Intuition in graph terms.** The spanning tree $\mathcal{T}$ contains no cycles, so collapsing it does not remove or merge any cycles in $G$. Each fundamental cycle in $G$ (formed by a non-tree edge and the unique tree path connecting its endpoints) is preserved in $G/\mathcal{T}$ as a loop based at the collapsed tree vertex. Therefore, the first homology group $H_1(G)$ — which measures independent cycles — remains unchanged. □

**Proposition 2.** *Each non-tree edge $e \in E \setminus \mathcal{T}$ induces a unique fundamental cycle in $G$, which becomes a nontrivial loop in $G/\mathcal{T}$. The collection of these loops forms a basis of the first homology group $H_1(G; \mathbb{Z})$, capturing all independent cycles and providing a concise topological summary of the graph.*

*Proof.* Let $G = (V, E)$ be a finite connected graph and $\mathcal{T} \subset G$ a spanning tree.

**Existence and uniqueness of fundamental cycles.** For each non-tree edge $e = (u, v) \in E \backslash \mathcal{T}$, there exists a unique simple path $P_{\mathcal{T}}(u, v)$ in $\mathcal{T}$ connecting $u$ and $v$, by acyclicity of $\mathcal{T}$. Concatenating $e$ with $P_{\mathcal{T}}(u, v)$ defines a unique simple cycle $C_e = e \cup P_{\mathcal{T}}(u, v) \subset G$. Under the quotient map $\gamma : G \to G/\mathcal{T}$ that collapses $\mathcal{T}$ to a point, the tree-path $P_{\mathcal{T}}(u, v)$ is mapped to that point, so $C_e$ becomes a nontrivial loop in $G/\mathcal{T}$.

**Spanning and independence in homology.** The cyclomatic number of $G$ is $\beta_1(G) = |E| - |V| + 1$, which equals the number of non-tree edges. Thus there are exactly $\beta_1(G)$ fundamental cycles.

Any cycle in $G$ can be expressed as a linear combination of these fundamental cycles: traversing a cycle, each time a non-tree edge $e$ is encountered, the corresponding $C_e$ can be used to eliminate segments along $\mathcal{T}$.

These cycles are independent in $H_1(G)$, because under $\gamma$, each fundamental cycle maps to a distinct loop in $G/\mathcal{T}$, and loops around different edges of a wedge of circles are linearly independent in homology.

Hence the set $\{[C_e] \mid e \in E \setminus \mathcal{T}\}$ forms a basis of $H_1(G)$. Different choices of spanning tree yield different sets of fundamental cycles as edge sets, but the corresponding homology classes always span $H_1(G)$. Therefore, these loops capture all independent cyclic dependencies of $G$, providing a concise topological summary suitable for lifting $G$ into a higher-dimensional cell complex. $\square$

## 4.2 TOPOLOGY-AWARE SUBCOMPLEX RETRIEVAL

Given a query $x_q$, we first encode it into a $d$-dimensional embedding:

$$z_q = \text{LM}(x_q) \in \mathbb{R}^d. \tag{11}$$

To retrieve the most relevant cells, we compute the cosine similarity between $z_q$ and the embeddings of 0- and 1-cells:

$$\mathcal{X}_k^{(d)} = \text{TopK}_{x^d \in X^{(d)}} \cos(z_q, z_{x^d}^d), \tag{12}$$

where $d \in \{0, 1\}$ denotes the dimension, $z_{x^d}^d$ is the embeddings of $d$-cell $x^d$. These provide the top-k relevant 0- and 1-cells.

*Prize assignment.* Each selected 0-cell $x^0 \in X_k^{(0)}$ and 1-cell $x^1 \in X_k^{(1)}$ is assigned a descending prize according to its ranking:

$$\text{prize}(x^i) = \begin{cases} k - r, & \text{if } x^i \text{ is ranked } r\text{-th among top-}k \text{ cells,} \\ 0, & \text{otherwise,} \end{cases} \quad i = 0, 1. \tag{13}$$

For each 2-cell $x^2 \in X^{(2)}$, its prize is computed from the prizes of its boundary cells:

$$\text{prize}(x^2) = \sum_{d \in \{0,1\}} \sum_{x^d \in \partial_d x^2} \text{prize}(x^d) - \text{cost}(x^2), \tag{14}$$

where $\partial_d x^2$ denote the sets of boundary $d$-cells of $x^2$, and $\text{cost}(x^2) = |\partial_1 x^2| \cdot C_2$ penalizes larger faces with a tunable constant $C_2$. Top-$k$ 2-cells are selected based on this prize ranking, denoted as $\mathcal{X}_k^{(2)}$, ensuring that all selected 2-cells share boundary cells with the chosen 0- and 1-cells.

*Subcomplex selection.* The final subcomplex $X^*$ maximizes the total prize while controlling size:

$$X^* = \underset{\substack{X' \subseteq X, \\ X' \text{ connected}}}{\text{argmax}} \sum_{d \in \{0,1,2\}} \sum_{x^d \in X'^{(d)}} \text{prize}(x^d) - \text{cost}(X'), \tag{15}$$

where $\text{cost}(X')$ is a size-dependent penalty. The boundary consistency constraint ensures that any selected 2-cell $x^2 \in X^{*(2)}$ has all of its boundary 0- and 1-cells included in $X^{*(0)}$ and $X^{*(1)}$, preserving topological coherence.

The resulting topology-aware subcomplex selection problem can be seen as a generalization of Prize-Collecting Steiner Tree (PCST) problem (Bienstock et al., 1993) to higher-dimensional cell

complexes with multi-dimensional prizes and size-dependent penalties. We adopt a near-linear time approximation algorithm Hegde et al. (2015) to efficiently identify a near-optimal connected sub-complex $X^*$. This ensures that the final subcomplex captures the most query-relevant structures across all cell dimensions, while maintaining computational efficiency and topological validity.

## 4.3 MULTI-DIMENSIONAL TOPOLOGICAL REASONING

After retrieving the query-relevant subcomplex $X^* = X^{*(0)} \cup X^{*(1)} \cup X^{*(2)}$, we propagate semantic and relational information across different topological dimensions to enable structured reasoning over the enriched cell complex. We employ a two-stage message passing mechanism that leverages the multi-dimensional structure of the complex. In the first stage, information is propagated along the 1-skeleton, between 0-cells and 1-cells, over $L$ hops:

$$\boldsymbol{h}_x^l = \text{UPDATE}^l\Big(\boldsymbol{h}_x^l, m_{\mathcal{F}}^l(x), m_{\mathcal{C}}^l(x)\Big), \quad x \in X^{*(0)} \cup X^{*(1)}, \quad l = 1, \dots, L, \tag{16}$$

where $m_{\mathcal{F}}^l(x)$ aggregates messages from faces, and $m_{\mathcal{C}}^l(x)$ aggregates messages from cofaces:

$$
\begin{aligned}
m_{\mathcal{F}}^{l+1}(x) &= \text{AGG}_{y \in \mathcal{F}(x)} M_{\mathcal{F}}(\boldsymbol{h}_x^l, \boldsymbol{h}_y^l), \\
m_{\mathcal{C}}^{l+1}(x) &= \text{AGG}_{z \in \mathcal{C}(x)} M_{\mathcal{C}}(\boldsymbol{h}_x^l, \boldsymbol{h}_z^l),
\end{aligned}
\tag{17}
$$

with $\mathcal{F}(x)$ and $\mathcal{C}(x)$ denoting the sets of faces and cofaces of $x$.

In the second stage, cells of all dimensions exchange information with higher-dimensional neighbors to capture multi-dimensional topological context. For each cell $x \in X^*$, the representation is updated as

$$\boldsymbol{h}_x^{L+1} = \text{UPDATE}\Big(\boldsymbol{h}_x^L, m_{\mathcal{F}}^L(x), m_{\mathcal{C}}^L(x), m_{\uparrow}^{L+1}(x)\Big), \tag{18}$$

where $m_{\uparrow}^{L+1}(x)$ aggregates messages from adjacent cells via shared cofaces. Specifically, the messages are defined as

$$m_{\uparrow}^{L+1}(x) = \text{AGG}_{w \in \mathcal{N}_{\uparrow}(x)} M_{\uparrow}(\boldsymbol{h}_x^L, \boldsymbol{h}_w^L, \boldsymbol{h}_{x \cup w}^{L+1}), \tag{19}$$

with $\mathcal{N}_{\uparrow}(x)$ the set of cells adjacent to $x$ via a shared coface.

To generate a fixed-dimensional representation of the entire subcomplex, we aggregate the embeddings of all its cells:

$$\boldsymbol{h}_{X^*} = \text{POOL}\Big(\{\boldsymbol{h}_x^{L+1} \mid x \in X^{*(0)} \cup X^{*(1)} \cup X^{*(2)}\}\Big) \in \mathbb{R}^{d_s}, \tag{20}$$

where POOL can be implemented as mean pooling over the cell embeddings, and $d_s$ denotes the dimension of the resulting subcomplex representation. This aggregated embedding $\boldsymbol{h}_{X^*}$ encodes both the semantic attributes of individual cells and the multi-dimensional topological context of the subcomplex, serving as input to the *Cell Complex-Augmented Generation* module for query-guided answer generation.

## 4.4 CELL COMPLEX-AUGMENTED GENERATION

With the subcomplex embedding $\boldsymbol{h}_{X^*}$ obtained from the Multi-dimensional Topological Reasoning module, we integrate it into a pretrained LLM to guide query-aware answer generation. First, we align the subcomplex embedding to the LLM's hidden space via a multilayer perceptron (MLP):

$$\hat{\mathbf{h}}_{X^*} = \text{MLP}_{\phi}(\boldsymbol{h}_{X^*}) \in \mathbb{R}^{d_l}, \tag{21}$$

where $d_l$ is the hidden dimension of the LLM. The projected vector $\hat{\mathbf{h}}_{X^*}$ acts as a soft prompt, providing structured, topologically-informed guidance to the LLM.

To leverage the LLM's text reasoning capabilities, we also transform the retrieved subcomplex into a textualized format, denoted as $\text{textualize}(X^*)$, by flattening the textual attributes of all cells while preserving the structural hierarchy. Given a natural language query $x_q$, we concatenate it with the textualized subcomplex and feed it into the LLM's embedding layer:

$$\boldsymbol{h}_t = \text{TextEmbedder}([\text{textualize}(X^*); x_q]) \in \mathbb{R}^{L \times d_l}, \tag{22}$$

where $[\cdot;\cdot]$ denotes concatenation, $L$ is the number of tokens, and the TextEmbedder is a frozen pretrained LLM embedding layer.

The final answer $Y$ is generated autoregressively, conditioned on both the soft subcomplex prompt $\hat{\mathbf{h}}_{X^*}$ and the textual token embeddings $\boldsymbol{h}_t$:

$$p_{\theta,\phi}(Y \mid X^*, x_q) = \prod_{i=1}^{r} p_{\theta,\phi}(y_i \mid y_{<i}, [\hat{\mathbf{h}}_{X^*}; \boldsymbol{h}_t]), \tag{23}$$

where $\theta$ denotes the frozen LLM parameters and $\phi$ denotes the trainable parameters of the MLP and the subcomplex encoder. Gradients are backpropagated through $\hat{\mathbf{h}}_{X^*}$, enabling the subcomplex encoder to learn to generate embeddings that are optimally informative for downstream generation.

## 5 EXPERIMENTS

### 5.1 EXPERIMENT SETUP

**Datasets.** Following prior work He et al. (2024), we use three existing datasets to conduct experiments [1]: WebQSP Yih et al. (2016), ExplaGraphs Saha et al. (2021) and SceneGraphs Hudson & Manning (2019). These datasets are standardized into a uniform format suitable for graph question answering He et al. (2024), allowing consistent evaluation across diverse reasoning tasks. More details about these datasets are provided in Appendix C.

**Comparison Methods.** To evaluate the performance of TopoRAG, we consider three categories of baselines. We provide more details in Appendix D.

*Inference-only LLMs* directly answer questions using the textual graph as input, including zero-shot prompting, zero-shot Chain-of-Thought (Zero-CoT) Kojima et al. (2022), Build-a-Graph prompting (CoT-BAG) Wang et al. (2023), and KAPING Baek et al. (2023), a knowledge-augmented zero-shot approach; *Frozen LLMs with prompt tuning* keep model parameters fixed while optimizing the input prompt, including soft prompt tuning, GraphToken Perozzi et al. (2024), G-Retriever He et al. (2024) with a frozen LLM and SubgraphRAG Li et al. (2025); *Tuned LLMs* update model parameters using LoRA Hu et al. (2021), including standard LoRA fine-tuning and G-Retriever w/ LoRA He et al. (2024) combining retrieval augmentation with parameter-efficient tuning, GNN-RAG Mavromatis & Karypis (2025).

**Evaluation Metrics.** For ExplaGraphs and SceneGraphs, the performance is measured using Accuracy, which calculates the percentage of correctly predicted answers. For WebQSP, we use the Hit metric, which measures the percentage of queries for which at least one of the top returned answers is correct. This metric is particularly suitable for multi-hop reasoning tasks, where the model must traverse multiple hops in a knowledge graph to retrieve the correct answer.

**Experiment Settings.** All experiments are conducted on two A6000-48G GPUs. For retrieval, we set the top-$k$ for 0- and 1-cells to $k = 3$ on WebQSP; on SceneGraphs, we set $k = 3$ for 0-cells and $k = 5$ for 1-cells. For 2-cells, we sweep the top-$k$ over $k \in \{0, 1, 2, 3\}$. For reasoning, the number of layers is varied in $\{2, 3, 4, 5\}$, with a uniform dimensionality of 1024 across all layers (input, hidden, and output). For generation, we employ the Llama-2-7B model Touvron et al. (2023) as the large language model backbone. When fine-tuning with LoRA Hu et al. (2021), we set the rank $= 8$, alpha $= 16$, and dropout rate $= 0.05$; for prompt tuning, we use 10 virtual tokens. The maximum input length is set to 512 tokens, and the maximum number of generated tokens is set to 32. We adopt the AdamW optimizer Loshchilov & Hutter (2017) with a learning rate of $1 \times 10^{-5}$, a batch size of 8, and train for 10 epochs with early stopping (patience $= 2$).

### 5.2 EXPERIMENT RESULT

**Main Results.** As summarized in Table 1, our model consistently outperforms all baselines across datasets and configurations. We highlight three key findings:

---

[1]https://github.com/Snnzhao/TopoRAG

Table 1: Performance comparison across `ExplaGraphs`, `SceneGraphs`, and `WebQSP` datasets under different configurations. The bold numbers indicate that the improvement of our model over the baselines is statistically significant with (p-value < 0.01), and the best baseline results are underlined

.

| Setting | Method | ExplaGraphs | SceneGraphs | WebQSP |
|---|---|---|---|---|
| Inference-only | Zero-shot | 0.5650 | 0.3974 | 41.06 |
| | Zero-CoT | 0.5704 | 0.5260 | 51.30 |
| | CoT-BAG | 0.5794 | 0.5680 | 39.60 |
| | KAPING | 0.6227 | 0.4375 | 52.64 |
| Frozen LLM w/ PT | Prompt tuning | 0.5763 | 0.6341 | 48.34 |
| | GraphToken | 0.8508 | 0.4903 | 57.05 |
| | G-Retriever | 0.8516 | 0.8131 | 70.49 |
| | SubgraphRAG | 0.8535 | 0.8074 | 86.61 |
| | *TopoRAG* (**Ours**) | 0.8899 | 0.8362 | 87.10 |
| Tuned LLM | LoRA | 0.8538 | 0.7862 | 66.03 |
| | G-Retriever w/ LoRA | 0.8705 | 0.8683 | 73.79 |
| | GNN-RAG | 0.8466 | 0.8149 | 85.70 |
| | *TopoRAG* w/ LoRA (**Ours**) | **0.9151** | **0.8768** | **90.66** |

- *TopoRAG* **delivers the strongest overall performance.** Compared to the best baseline, *TopoRAG* improves `ExplaGraphs` and `SceneGraphs` Accuracy by 5.12% and 0.98%, respectively; on `WebQSP`, it increases the Hit metric by 4.67%. We attribute the improvements to the following reasons: 1) *Cellular Representation Lifting*, which transforms textual graphs into cellular complexes and explicitly encodes higher-dimensional structures that support closed-loop relational reasoning; 2) *topology-aware subcomplex retriever* that selects query-relevant cellular complexes, supplying compact yet informative topological context; and 3) *multi-dimensional topological reasoning* that propagates information across 0-/1-/2-cells to enable structured, logic-aware inference tightly integrated with LLM reasoning. Together, these components overcome the limitations of node/edge-centric methods and yield more accurate and robust QA over complex textual graphs.

- **Graph-structured prompts effectively improve QA performance.** All prompt-tuning approaches (e.g., GraphToken, SubgraphRAG) outperform inference-only baselines (Zero-shot, Zero-CoT), underscoring the value of structured context. *TopoRAG* further improves upon these by grounding prompts in higher-dimensional topological dependencies—beyond nodes and edges—thereby providing richer, loop-aware relational context, especially for queries involving multi-hop and cyclic dependencies.

**Ablation Study.** We conduct an ablation study to evaluate the contribution of each component of TopoRAG. Specifically, we replace *Cellular Representation Lifting* (**CRL**) with a standard edge-based graph structure, substitute *Topology-aware Subcomplex Retrieval* (**TSR**) with shortest-path-based retrieval, and replace *Multi-dimensional Topological Reasoning* (**MTR**) with a GCN for message passing.

Table 2: Ablation Study on ExplaGraphs and WebQSP Datasets.

| Method | ExplaGraphs (Accuracy) | WebQSP (Hit) |
|---|---|---|
| w/o CRL | 0.8576 | 84.96 |
| w/o TSR | 0.8524 | 84.23 |
| w/o MTR | 0.8611 | 85.46 |
| **TopoRAG** | **0.9151** | **90.66** |

1) Replacing *Cellular Representation Lifting* (**CRL**) with an edge-only graph representation removes the lifting of textual graphs into cellular complexes and, consequently, the explicit encoding of higher-dimensional structures (e.g., cycles) that support closed-loop relational reasoning. This leads to a substantial performance drop, underscoring the necessity of CRL for modeling high-dimensional dependencies in RAG.

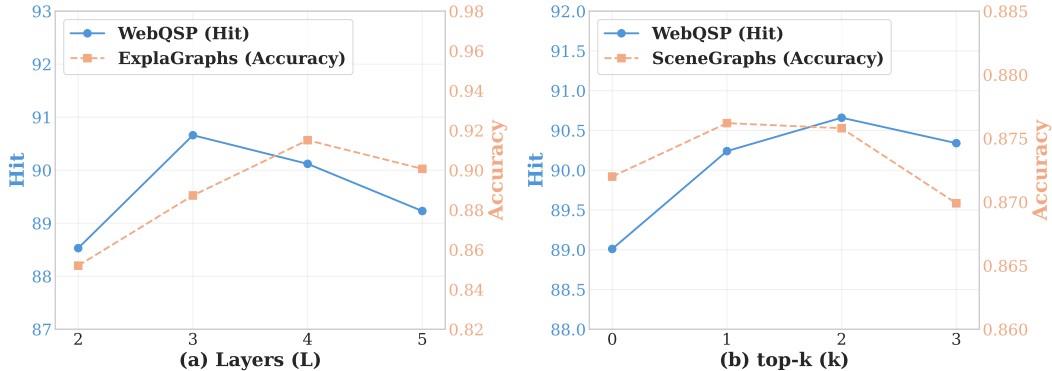

Figure 3: (a) Effect of layers $L \in \{2, 3, 4, 5\}$ on *TopoRAG* performance: WebQSP (Hit) and Expla-Graphs (Accuracy). (b) Effect of top-$k$ $k \in \{0, 1, 2, 3\}$ on *TopoRAG* performance: WebQSP (Hit) and SceneGraphs (Accuracy).

2) Replacing *Topology-aware Subcomplex Retrieval* (TSR) with shortest-path retrieval restricts results to the 1-skeleton and removes 2-cells, thereby discarding higher-dimensional motifs and closed-loop constraints that are critical for capturing query-relevant structure, underscoring the importance of TSR for sustaining *TopoRAG*'s effectiveness.

3) When removing *Multi-dimensional Topological Reasoning* (**MTR**), the performance of TopoRAG drops significantly. This is due to the crucial role of MTR in enabling multi-dimensional message passing using cellular complexes. Without MTR, the model loses the ability to effectively propagate information from high-dimensional cells to low-dimensional ones, resulting in the loss of important high-dimensional structural information during the message passing process.

**Hyper-parameter Study.** We study the sensitivity of TopoRAG to two key hyperparameters: the number of layers $L$ and the top-$k$ of 2-cell for subcomplex retrieval. The layer depth $L$ controls the model's ability to capture hierarchical structures and long-range dependencies. Larger $L$ enhances the model's representational capacity but may lead to overfitting or increased computational cost, while smaller $L$ may limit structural information capture, causing underfitting. Figure 3 (a) shows the effect of different $L$ values on performance: reasoning ability improves as $L$ increases, but excessive depth reduces expressiveness. We also analyze the impact of the top-$k$ parameter for 2-cells selection on retrieval. Too small $k$ causes information loss, while too large $k$ introduces noise. Figure 3 (b) illustrates the effect of different $k \in \{0, 1, 2, 3\}$ values, showing that a moderate $k$ achieves a better balance between structural coverage and noise. In Appendix F, we present an extended sensitivity analysis of the choice of $k$.

## 6  CONCLUSION

In this work, we introduced **TopoRAG**, a topology-enhanced retrieval-augmented generation framework for textual graph question answering. Unlike conventional GraphRAG approaches that mainly rely on nodes and edges, TopoRAG explicitly incorporates higher-dimensional topological structures by lifting textual graphs into cellular complexes. Through a topology-aware subcomplex retrieval mechanism, TopoRAG provides compact yet informative multi-dimensional contexts, while the proposed multi-dimensional topological reasoning module enables structured and logic-aware inference that captures cyclic and higher-dimensional dependencies beyond pairwise relations. Experimental results demonstrate that TopoRAG outperforms existing methods across three datasets from different domains.

### ACKNOWLEDGMENTS

This work is supported in part by the Chongqing Municipal Education Commission (No. KJQN202500620), and General Program of Chongqing Municipal Fund (NO. CSTB2025NSCQ-

GPX1297). The authors would like to thank the anonymous reviewers for their valuable comments and advice.

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

Table 3: Notation and definitions used in the TopoRAG framework.

| Notation | Definition |
|---|---|
| $\mathcal{G}$ | The input textual graph, defined as $\mathcal{G} = (V, E, \{t_n\}_{n \in V}, \{t_e\}_{e \in E})$. |
| $V, E$ | The sets of vertices (nodes) and edges in the graph $\mathcal{G}$. |
| $t_n, t_e$ | The text attributes associated with node $n \in V$ and edge $e \in E$. |
| $X$ | The regular cell complex constructed from the graph $\mathcal{G}$. |
| $X^{(d)}$ | $d$-skeleton. |
| $x^k$ | A $k$-dimensional cell in the complex (e.g., $x^0$: a 0-cell, $x^1$: a 1-cell). |
| $z_v^0, z_e^1$ | The $d$-dimensional embedding of node $v$ (0-cell) and edge $e$ (1-cell), obtained via a Language Model (LM). |
| $\mathcal{T}$ | A spanning tree of the graph $\mathcal{G}$, used for cycle detection. |
| $x_e^2$ | A 2-cell attached to a fundamental cycle induced by a non-tree edge $e \in E \setminus \mathcal{T}$. |
| $z_q$ | The $d$-dimensional embedding of the input query $x_q$. |
| $\mathcal{X}_k^{(d)}$ | The set of top-$k$ retrieved $d$-cells ($d = 0, 1, 2$) based on semantic similarity or prize. |
| $\text{prize}(x^i)$ | The prize (relevance score) assigned to cell $x^i$ during retrieval. |
| $X^*$ | The final retrieved and connected subcomplex, $X^* = X^{(0)*} \cup X^{(1)*} \cup X^{(2)*}$. |
| $\partial_k x$ | The boundary operator; $\partial_k x$ gives the set of $(k-1)$-cells on the boundary of a $k$-cell $x$. |
| $\boldsymbol{h}_x^l$ | The hidden representation of cell $x$ at message passing layer $l$. |
| $\mathcal{F}(x), \mathcal{C}(x)$ | The set of faces (lower-dimensional boundary cells) and cofaces (higher-dimensional incident cells) of cell $x$. |
| $m_{\mathcal{F}}^l(x), m_{\mathcal{C}}^l(x)$ | Messages aggregated from faces and cofaces of cell $x$ at layer $l$. |
| $\boldsymbol{h}_{X^*}$ | The final pooled representation of the entire subcomplex $X^*$. |
| $\hat{\mathbf{h}}_{X^*}$ | The projected subcomplex embedding, aligned to the LLM's hidden space via an MLP. |
| $p_{\theta,\phi}(Y \mid X^*, x_q)$ | The conditional probability of generating answer $Y$, given the subcomplex $X^*$ and query $x_q$. |

## A NOTATIONS

The notations in the TopoRAG framework are summarized in Table 3.

## B ADDITIONAL RELATED WORK

**Graphs and Large Language Models.** In parallel, there has been a surge of interest in combining graphs with LLMs Pan et al. (2023); Li et al. (2023b); Jin et al. (2023); Wang et al. (2023); Zhang et al. (2023). This line of research spans a wide spectrum, from the design of general graph models Ye et al. (2023); Liu et al. (2023); Yu et al. (2023b); Lei et al. (2023); Tang et al. (2023); Perozzi et al. (2024), to multi-modal architectures Li et al. (2023a); Yoon et al. (2023), and diverse downstream applications.

Applications of Graph-augmented LLMs include fundamental graph reasoning Zhang (2023); Chai et al. (2023); Zhao et al. (2023b), node classification He et al. (2023); Huang et al. (2023a); Sun et al. (2023); Chen et al.; Yu et al. (2023a); Chen et al. (2024); Qin et al. (2023), and graph classification/regression Qian et al. (2023); Zhao et al. (2023a). Furthermore, LLMs have been increasingly employed for knowledge graph-related tasks such as reasoning, completion, and question answering Tian et al. (2023); Jiang et al. (2023); Luo et al. (2023).

**Topological deep learning.** Topological deep learning expands graph learning by modelling relations that exceed simple pairwise links. Early work in Topological Signal Processing (TSP) emphasized the value of higher-dimensional structure for signal and relational modeling Barbarossa & Sardellitti (2020); Schaub et al. (2021); Roddenberry et al. (2022); Sardellitti et al. (2021), prompting extensions of graph tools to richer discrete geometries such as simplicial and cell complexes. Theoretical progress—e.g., higher-dimensional generalizations of the Weisfeiler–Lehman test—has clar-

ified the expressive power required for distinguishing complex topologies and motivated message-passing schemes beyond traditional GNNs Bodnar et al. (2021c;a).

On the modelling side, researchers have proposed numerous neural architectures that operate on these higher-dimensional domains, including convolutional-style operators for simplicial and cell complexes Ebli et al. (2020); Yang et al. (2022); Hajij et al. (2020); Yang & Isufi (2023); Roddenberry et al. (2021); Hajij et al. (2022) and attention-based formulations that incorporate incidence relations and coface interactions Goh et al.; Giusti et al. (2023). Efforts to unify these variants led to combinatorial-complex frameworks that generalize message passing to a wide class of combinatorial objects (simplicial complexes, CW complexes, hypergraphs) Hajij et al. (2023). Complementary lines of work use sheaf-theoretic constructions to impose local consistency and handle heterophilous patterns, demonstrating another principled route to encode localized topological constraints Hansen & Ghrist (2019); Hansen & Gebhart (2020); Bodnar et al. (2022); Battiloro et al. (2023; 2024); Barbero et al. (2022).

While these works have substantially advanced higher-dimensional representation learning, current topological deep learning methods exhibit limited generalization to unseen graphs with substantially different topologies, a sensitivity that hinders their applicability in real-world scenarios where structural variability is common. Moreover, they remain largely tied to static, pre-defined complexes and traditional graph formulations, leaving them ill-suited for reasoning in question answering tasks that require integrating textual graph inputs with large language models and dynamically incorporating external knowledge. In this work, we address these limitations by introducing TopoRAG, a framework that explicitly models multi-dimensional topological structures and couples topology-aware retrieval with large language models to enable robust and context-sensitive reasoning over textual graphs.

## C  DATASETS

Following prior work He et al. (2024), we use three existing datasets: `WebQSP`, `ExplaGraphs` and `SceneGraphs`, which are summarized in Table 4. These datasets are standardized into a uniform format suitable for graph question answering, allowing consistent evaluation across diverse reasoning tasks. `ExplaGraphs` focuses on generative commonsense reasoning. The task requires predicting whether an argument supports or contradicts a given belief, evaluated using Accuracy. `SceneGraphs` is a visual question answering dataset. The task is to answer questions based on textualized scene graphs, requiring spatial reasoning and multi-step inference, evaluated by Accuracy. `WebQSP` is a large-scale multi-hop knowledge graph QA dataset, which contains facts within 2-hops of entities mentioned in the questions. Each question is associated with a subgraph extracted from Freebase. The task involves multi-hop reasoning, evaluated using Hit for the top returned answer.

Table 4: Overview of datasets used in the GraphQA benchmark.

| Dataset | ExplaGraphs | SceneGraphs | WebQSP |
|---|---|---|---|
| Number of Graphs | 2,766 | 100,000 | 4,737 |
| Average Nodes | 5.17 | 19.13 | 1,370.89 |
| Average Edges | 4.25 | 68.44 | 4,252.37 |
| Node Features | Commonsense concepts | Object properties (*e.g.,* color, shape) | Freebase entities |
| Edge Features | Commonsense relations | Object interactions and spatial relations | Freebase relations |
| Task Type | Commonsense reasoning | Scene graph QA | Knowledge graph QA |
| Evaluation Metric | Accuracy | Accuracy | Hit |

## D  COMPARISON METHODS

In our experiments, we consider three categories of baselines: 1) *Inference-only*, 2) *Frozen LLM w/ prompt tuning (PT)*, 3) *Tuned LLM*. The details of each baseline are described as follows.

**Inference-only.**  Using a frozen LLM for direct question answering with textual graph and question.

- Zero-shot. In this approach, the model is given a textual graph description and a task description, and is immediately asked to produce the desired output. No additional examples or demonstrations are provided.

- Zero-CoT. Zero-shot Chain-of-thought (Zero-CoT) prompting (Kojima et al., 2022) is a follow-up to CoT prompting (Wei et al., 2022), which introduces an incredibly simple zero shot prompt by appending the words "Let's think step by step." to the end of a question.

- CoT-BAG. Build-a-Graph Prompting (BAG) (Wang et al., 2023) is a prompting technique that adds "Let's construct a graph with the nodes and edges first." after the textual description of the graph is explicitly given.

- KAPING. KAPING (Baek et al., 2023) is a zero-shot knowledge-augmented prompting method for knowledge graph question answering. It first retrieves triples related to the question from the graph, then prepends them to the input question in the form of a prompt, which is then forwarded to LLMs to generate the answer.

**Frozen LLM w/ prompt tuning (PT).** Keeping the parameters of the LLM frozen and adapting only the prompt.

- GraphToken (Perozzi et al., 2024), which is a graph prompt tuning method.

- G-Retriever He et al. (2024) is an efficient and lightweight model that adapts frozen large language model parameters to graph question answering tasks solely through trainable graph-structured soft prompts.

- SubgraphRAG Li et al. (2025) is a retrieval-augmented generation framework based on knowledge graphs, which effectively improves the accuracy, efficiency, and interpretability of question answering through lightweight subgraph retrieval and inference with an untuned large language model.

**Tuned LLM.** Fine-tuning the LLM with LoRA.

- G-Retriever (w/ LoRA) He et al. (2024) is a high-precision model that fine-tunes large language model parameters using techniques such as LoRA, enabling deep integration of graph structure information to enhance graph question answering performance.

- GNN-RAG Mavromatis & Karypis (2025) is a retrieval-augmented generation framework based on graph neural networks, which efficiently retrieves multi-hop reasoning paths from knowledge graphs using GNNs and inputs them as context to an LLM, enhancing the accuracy and efficiency of complex knowledge graph question answering.

## E IMPACT OF DIFFERENT SPANNING TREE METHODS

We evaluate the sensitivity of our method to the choice of spanning tree by comparing different spanning tree construction strategies, including DFS, BFS, and Random spanning trees. As shown in Table 5, all variants achieve comparable performance across datasets, indicating that the proposed framework is largely insensitive to the specific spanning tree algorithm.

This robustness can be explained from a topological perspective. Although different spanning trees induce different fundamental cycle bases, they generate the same cycle space, corresponding to an identical first homology group. Consequently, while the specific 2-cells constructed from basis cycles may vary, the lifted complexes preserve the same global topological structure.

Table 5: Performance comparison of different spanning tree algorithms

| Method | ExplaGraphs (Accuracy) | SceneGraphs (Accuracy) | WebQSP (Hit) |
|--------|------------------------|------------------------|--------------|
| DFS    | 0.9151                 | 0.8768                 | 90.66        |
| BFS    | 0.9025                 | 0.8725                 | 90.18        |
| Random | 0.9187                 | 0.8698                 | 90.03        |

## F    IMPACT OF TOP-$K$ SELECTION ON SUBCOMPLEX RETRIEVAL

To further quantify the influence of $k$ on subcomplex retrieval, we conduct supplementary experiments with $k \in \{1, 2, 3\}$ and report, on `WebQSP` and `SceneGraphs`, the average numbers of 0-, 1-, and 2-cells per retrieved subcomplex. As shown in Table 6, larger $k$ yields more retrieved 2-cells; the accompanying inclusion of higher–dimensional structures also increases the counts of 0- and 1-cells, underscoring the trade-off that $k$ introduces between structural coverage and noise.

Table 6: Impact of Top-$K$ Selection on Subcomplex Retrieval.

| Dataset | $k$ | Number of Cells per Dimension | | |
|---|---|---|---|---|
| | | 0-cells | 1-cells | 2-cells |
| `WebQSP` | 1 | 15 | 17 | 2 |
| | 2 | 16 | 18 | 3 |
| | 3 | 17 | 20 | 4 |
| `SceneGraphs` | 1 | 9 | 12 | 4 |
| | 2 | 9 | 13 | 5 |
| | 3 | 9 | 14 | 6 |

## G    DISCUSSION ON THE COMPLEXITY

Following prior work, TopoRAG adopts the LLM+X framework, which enhances LLMs with multimodal capabilities by integrating them with encoders from other modalities. In recent years, the LLM+X framework has been widely adopted in various RAG methods. Notable examples include: 1) KG+LLM approaches, such as RoG, ToG, StructGPT, and KAPING, and 2) GNN+LLM approaches, including G-Retriever and GNN-RAG. These models have demonstrated strong performance in both efficiency and accuracy. Furthermore, we introduce Topo+LLM.

Regarding the integration of topological methods into RAG, it does not significantly increase the time or computational complexity associated with LLM-based answer generation, as both cellular complex construction and subcomplex retrieval are implemented during the preprocessing phase.

In contrast to G-Retriever, we use a more complex *Multi-dimensional Topological Reasoning* to capture high-dimensional topological structures, which introduces some additional time overhead. However, the benefits achieved outweigh the costs. To validate this, we conducted experiments using two A6000-48G GPUs with Llama2-7b as the LLM, training on `ExplaGraphs` and `SceneGraphs`. Detailed experimental settings are provided in Appendix C. The results in Table 7 show that, compared to baseline methods, our approach incurs only a slight increase in runtime, yet significantly improves model performance.

Table 7: Performance and Efficiency Comparison of TopoRAG and G-Retriever on `ExplaGraphs` and `SceneGraphs` Datasets.

| Setting | Method | ExplaGraphs | | SceneGraphs | |
|---|---|---|---|---|---|
| | | Hit | Time | Accuracy | Time |
| Frozen LLM w/ PT | G-Retriever | 0.8516 | 2.6 min/epoch | 0.8131 | 267 min/epoch |
| | *TopoRAG* | 0.8899 | 3.0 min/epoch | 0.8362 | 300 min/epoch |
| Tuned LLM | G-Retriever w/ LoRA | 0.8705 | 3.0 min/epoch | 0.8683 | 285 min/epoch |
| | *TopoRAG* w/ LoRA | 0.9151 | 3.3 min/epoch | 0.8768 | 310 min/epoch |

# H   ALGORITHMS

**Cellular Representation Lifting Algorithm.**   We present Algorithm 1, which formally describes the Cellular Representation Lifting process outlined in Section 4.1. The algorithm takes a textual graph $\mathcal{G}$ and lifts it into a regular cell complex $X$ endowed with feature representations for all its cellular substructures. The algorithm proceeds in two main phases.

The first phase (Lines 1–15) constructs the **1-skeleton** ($X^{(1)}$) of the complex. It initializes the sets of 0-cells ($X^{(0)}$) and 1-cells ($X^{(1)}$) and their corresponding feature dictionaries ($\mathbf{Z}^0$, $\mathbf{Z}^1$). For each vertex $v \in V$, a 0-cell $x_v^0$ is instantiated, and its representation $z_v^0$ is obtained by encoding the vertex's text attribute $t_v$ using a pre-trained language model (LM). Similarly, for each edge $e \in E$, a 1-cell $x_e^1$ is created, attached to the 0-cells of its endpoints, and its representation $z_e^1$ is computed from the edge text $t_e$.

The second phase (Lines 16–28) augments the 1-skeleton with **2-cells** to capture higher-dimensional topological information. A spanning tree $\mathcal{T}$ of $\mathcal{G}$ is computed, whose complement $E_{\text{non-tree}}$ defines the set of fundamental cycles in the graph. For each non-tree edge $e \in E_{\text{non-tree}}$, the algorithm identifies the corresponding fundamental cycle by finding the unique path between the endpoints of $e$ in $\mathcal{T}$ and appending $e$ itself (as detailed in the subroutine Algorithm 2). A 2-cell $x_e^2$ is then created and attached to this cycle. The initial representation $z_e^2$ for the 2-cell is computed by aggregating (e.g., via mean pooling) the representations of all the 0-cells and 1-cells that constitute its boundary cycle. This provides an inductive bias that initializes the feature of a higher-dimensional cavity based on the features of its lower-dimensional boundaries.

Finally, the complete cell complex $X$ is formed by the union of all cells across dimensions (Line 30). The algorithm returns both the topological structure $X$ and the associated features $\mathbf{Z}^0, \mathbf{Z}^1, \mathbf{Z}^2$, which serve as the input for subsequent cellular message-passing networks (Bodnar et al., 2021b). This lifting procedure effectively transforms a plain graph into a richer topological domain, explicitly encoding relational cycles as tangible geometric entities.

**Topology-aware Subcomplex Retrieval Algorithm.**   Algorithm 3 formalizes the Topology-aware Subcomplex Retrieval process outlined in Section 4.2. The algorithm takes as input the cell complex $X$ with precomputed cellular embeddings, a textual query $x_q$, and parameters $k$ and $C_2$. It returns a connected subcomplex $X^*$ that maximizes the relevance prize under topological constraints. The procedure operates in three distinct phases.

The first Phase is Query-based Cell Retrieval (Lines 1–7). The algorithm begins by encoding the query $x_q$ into an embedding $z_q$ using the language model (LM). It then computes the cosine similarity between $z_q$ and the embeddings of all 0-cells and 1-cells. The top-$k$ most similar cells from each dimension are selected, forming the initial sets of highly relevant candidates, $X_k^{(0)}$ and $X_k^{(1)}$.

The second Phase is Multi-dimensional Prize Assignment (Lines 9–26). This phase assigns a prize value to each cell, quantifying its incentive for inclusion in the final subcomplex. Prizes for the top-$k$ 0-cells and 1-cells are assigned in descending order based on their similarity ranking (e.g., the highest-ranked cell receives a prize of $k$). For a 2-cell $x^2$, its prize is computed inductively as the sum of the prizes of all its boundary 0-cells and 1-cells, minus a cost penalty $\text{cost}(x^2) = |\partial_1 x^2| \cdot C_2$ that discourages the selection of overly large faces. This design propagates relevance signals from lower-dimensional cells to the higher-dimensional structures they define. Finally, the top-$k$ 2-cells by prize, denoted $X_k^{(2)}$, are selected.

The last phase is Prize-Collecting Steiner Subcomplex Extraction (Lines 28–34). The core challenge is to extract a *connected* subcomplex that includes high-prize cells while respecting the *boundary consistency constraints* (i.e., if a 2-cell is selected, all its boundary cells must also be included). We model this as a Prize-Collecting Steiner Tree (PCST) problem on a hypergraph representation $G_{\text{hyper}}$ of the cell complex, where 2-cells are modeled as hyperedges. The set $R_{\text{terminals}}$ consists of the top-$k$ cells from all dimensions, and the prize function $\mathcal{P}$ is defined from the previous phase. Solving this generalized PCST problem yields a connected subcomplex $X^*$ that approximates the optimal trade-off between total collected prize and the cost of the required connecting cells. We employ a near-linear time approximation algorithm (Hegde et al., 2015) to ensure computational feasibility.

The algorithm's output, $X^*$, provides a coherent, query-focused topological summary of the original complex, which can be directly utilized for downstream tasks such as reasoning or explanation generation.

---

**Algorithm 1** Cellular Lifting of Textual Graphs

---

**Require:** Textual graph $\mathcal{G} = (V, E, \{t_v\}_{v \in V}, \{t_e\}_{e \in E})$, pre-trained language model $\text{LM}(\cdot)$.
**Ensure:** A regular cell complex $X$ with cellular embeddings $\{z_v^0\}, \{z_e^1\}, \{z_e^2\}$.
 1: **Step 1: Construct the 1-Skeleton and Compute Initial Embeddings**
 2: $X^{(0)} \leftarrow \emptyset, X^{(1)} \leftarrow \emptyset$ {Initialize sets of 0-cells and 1-cells}
 3: $\mathbf{Z}^0 \leftarrow \{\}, \mathbf{Z}^1 \leftarrow \{\}$ {Initialize dictionaries for embeddings}
 4: **for** $v \in V$ **do**
 5:     Create a 0-cell $x_v^0$ for vertex $v$
 6:     $X^{(0)} \leftarrow X^{(0)} \cup \{x_v^0\}$
 7:     $z_v^0 \leftarrow \text{LM}(t_v)$ {Encode vertex text attribute}
 8:     $\mathbf{Z}^0[x_v^0] \leftarrow z_v^0$
 9: **end for**
10: **for** $e = (u, v) \in E$ **do**
11:     Create a 1-cell $x_e^1$ attached to $x_u^0$ and $x_v^0$
12:     $X^{(1)} \leftarrow X^{(1)} \cup \{x_e^1\}$
13:     $z_e^1 \leftarrow \text{LM}(t_e)$ {Encode edge text attribute}
14:     $\mathbf{Z}^1[x_e^1] \leftarrow z_e^1$
15: **end for**
16: $X^{(1)} \leftarrow X^{(0)} \cup X^{(1)}$ {The 1-skeleton is complete}
17:
18: **Step 2: Augment with 2-Cells to Capture Higher-Dimensional Structures**
19: $X^{(2)} \leftarrow \emptyset$
20: $\mathbf{Z}^2 \leftarrow \{\}$
21: $\mathcal{T} \leftarrow \text{SpanningTree}(\mathcal{G})$ {e.g., using BFS or DFS}
22: $E_{\text{non-tree}} \leftarrow E \setminus \mathcal{T}$
23: **for** $e \in E_{\text{non-tree}}$ **do**
24:     $u, v \leftarrow \text{endpoints}(e)$
25:     $\text{cycle} \leftarrow \text{FindFundamentalCycle}(e, \mathcal{T})$
26:     {cycle is the unique path from $u$ to $v$ in $\mathcal{T}$ plus edge $e$}
27:     Create a 2-cell $x_e^2$
28:     Attach $x_e^2$ to the 1-skeleton via the attaching map $\varphi_e : \partial D^2 \rightarrow \text{cycle}$
29:     $X^{(2)} \leftarrow X^{(2)} \cup \{x_e^2\}$
30:     $z_e^2 \leftarrow \text{AggregateCycleEmbeddings}(\text{cycle}, \mathbf{Z}^0, \mathbf{Z}^1)$
31:     {e.g., mean/max pooling of embeddings of all 0/1-cells in the cycle}
32:     $\mathbf{Z}^2[x_e^2] \leftarrow z_e^2$
33: **end for**
34:
35: $X \leftarrow X^{(0)} \cup X^{(1)} \cup X^{(2)}$ {The final cell complex}
36:
37: **return** $X, \mathbf{Z}^0, \mathbf{Z}^1, \mathbf{Z}^2$

---

**Algorithm 2** Find Fundamental Cycle

---

**Require:** Non-tree edge $e = (u, v)$, spanning tree $\mathcal{T}$.
**Ensure:** An ordered list of 0-cells and 1-cells forming the fundamental cycle.
 1: $\text{path\_u\_to\_v} \leftarrow \text{GetUniquePathInTree}(u, v, \mathcal{T})$
 2: $\text{cycle\_vertices} \leftarrow \text{path\_u\_to\_v.vertices}$
 3: $\text{cycle\_edges} \leftarrow \text{path\_u\_to\_v.edges}$
 4: $\text{cycle\_edges.append}(e)$ {Add the non-tree edge to complete the cycle}
 5:
 6: **return** cycle\_vertices, cycle\_edges

---

---

**Algorithm 3** Topology-Aware Subcomplex Retrieval

---

**Require:** Cell complex $X = X^{(0)} \cup X^{(1)} \cup X^{(2)}$ with embeddings $\mathbf{Z}^0, \mathbf{Z}^1, \mathbf{Z}^2$; query $x_q$; parameters $k, C_2$.
**Ensure:** A connected, topology-aware subcomplex $X^* \subseteq X$.
 1: **Step 1: Encode Query and Retrieve Top-$k$ Cells**
 2: $z_q \leftarrow \text{LM}(x_q)$ {Encode the query}
 3: $S^0 \leftarrow \text{ComputeCosineSimilarity}(z_q, \mathbf{Z}^0)$ {$S^0$ is a list of (cell, score) pairs for all 0-cells}
 4: $S^1 \leftarrow \text{ComputeCosineSimilarity}(z_q, \mathbf{Z}^1)$ {$S^1$ is a list of (cell, score) pairs for all 1-cells}
 5: $X_k^{(0)} \leftarrow \text{ArgTopK}(S^0, k)$ {Set of top-$k$ relevant 0-cells}
 6: $X_k^{(1)} \leftarrow \text{ArgTopK}(S^1, k)$ {Set of top-$k$ relevant 1-cells}
 7:
 8: **Step 2: Prize Assignment**
 9: $\mathcal{P} \leftarrow \{\}$ {Initialize a prize dictionary for all cells}
10: **// Assign prizes to top-k 0/1-cells based on ranking**
11: $\text{rank} \leftarrow 0$
12: **for** $x^0 \in X_k^{(0)}$ (in descending order of similarity) **do**
13:     $\mathcal{P}[x^0] \leftarrow k - \text{rank}$
14:     $\text{rank} \leftarrow \text{rank} + 1$
15: **end for**
16: $\text{rank} \leftarrow 0$
17: **for** $x^1 \in X_k^{(1)}$ (in descending order of similarity) **do**
18:     $\mathcal{P}[x^1] \leftarrow k - \text{rank}$
19:     $\text{rank} \leftarrow \text{rank} + 1$
20: **end for**
21: **// Compute prizes for 2-cells based on boundary cells**
22: **for** $x^2 \in X^{(2)}$ **do**
23:     $\text{boundary\_prize} \leftarrow 0$
24:     **for each** $x^0 \in \partial_0 x^2$ **do** $\text{boundary\_prize} \leftarrow \text{boundary\_prize} + \mathcal{P}.get(x^0, 0)$
25:     **for each** $x^1 \in \partial_1 x^2$ **do** $\text{boundary\_prize} \leftarrow \text{boundary\_prize} + \mathcal{P}.get(x^1, 0)$
26:     $\text{cost} \leftarrow |\partial_1 x^2| \cdot C_2$ {Penalize larger faces}
27:     $\mathcal{P}[x^2] \leftarrow \text{boundary\_prize} - \text{cost}$
28: **end for**
29: $X_k^{(2)} \leftarrow \text{ArgTopK}(\{\mathcal{P}[x^2] \text{ for } x^2 \in X^{(2)}\}, k)$ {Select top-$k$ 2-cells by prize}
30:
31: **Step 3: Prize-Collecting Subcomplex Selection**
32: $V \leftarrow X^{(0)} \cup X^{(1)} \cup X^{(2)}$ {Candidate cells across all dimensions}
33: $R \leftarrow X_k^{(0)} \cup X_k^{(1)} \cup X_k^{(2)}$ {Top-$k$ cells as prize terminals}
34: **// Assign prizes to 0- and 1-cells based on query similarity**
35: **// Compute 2-cell prizes from boundary consistency with selected cells**
36:
37: $X^* \leftarrow \text{ApproxPCSTComplex}(X, R, \mathcal{P})$
38:     {Solve generalized PCST over cell complex with multi-dimensional prizes}
39: **// Enforce boundary consistency: any $2$-cell in $X^*$ must share boundary with chosen $0$- and $1$-cells**
40:
41: **return** $X^*$

---

## I STATEMENT

**Ethics Statement.** This research follows the ICLR Code of Ethics. The work does not involve experiments with human participants, collection of personal data, or sensitive information. All datasets employed are publicly released and widely used in prior studies. The proposed framework is intended to advance topology-aware retrieval and reasoning in textual graphs for question answering, and we do not anticipate direct societal risks or harmful misuse. Potential biases have been carefully considered, and we adopt standard practices to minimize unintended artifacts. No external funding sources or conflicts of interest influenced the conduct of this work.

**Reproducibility Statement.** To support reproducibility, all datasets are publicly available and preprocessing procedures are documented in the main text or supplementary material. We provide a complete implementation as part of the supplementary package, which enables replication of our experiments without additional dependencies. Hyperparameters, training configurations, and evaluation protocols are described in detail to facilitate verification and future extensions.

**LLM Usage.** We acknowledge the use of large language models (LLMs) as auxiliary tools for improving readability and clarity of writing. The conceptual development, methodology design, experimental setup, and analysis were entirely conducted and validated by the authors. LLMs were not used to generate novel research ideas nor to contribute to technical content.

