# OpenReview forum: "Topology of Reasoning: Retrieved Cell Complex-Augmented Generation for Textual Graph Question Answering"
_ICLR.cc/2026/Conference — ICLR 2026 Poster_

### Official Review · Reviewer_gvvE · 2025-10-27

**Soundness:** 4
**Presentation:** 4
**Contribution:** 3
**Rating:** 6
**Confidence:** 4

**Summary:**

The paper introduces a topology-aware retrieval-augmented generation framework that represents textual graphs as cell complexes to capture higher-order relations such as cycles and loops beyond standard nodes and edges. It retrieves relevant multi-dimensional subcomplexes and performs topological reasoning across dimensions to guide an LLM in generating answers.

**Strengths:**

- A novel approach to include more relevant higher dimensional information in the RAG pipeline
- Mathematically strong paper.
- The approach does improve upon the existing baselines.
- The approach provides interpretable reasoning traces

**Weaknesses:**

- The computational and memory cost of building and reasoning over high-dimensional cell complexes is not quantified or compared with simpler baselines.
- The approach still has its limitations - including more higher dimensional knowledge in case of incomplete/noisy graphs could lead to worse results. Moreover the approach will be very difficult to adapt in cases of dynamic graphs where new nodes/edges are added from time to time.

**Questions:**

- Can the topological retrieval appriach be adapted to handle multimodal graphs?
- Could the proposed topological framework be used for reasoning tasks that do not explicitly involve graph structures  (eg. scientific question answering or commonsense reasoning)?
- What is the intuition behind describing the graph as a cellular representation?
- Instead of adding high dimensional knowledge, can't we instead find the relevant 0- and 1- dim and let any reasoning model to reason over this? My intuition is that doing this will have many benefits - limited training, more robust to graph changes etc. Although this won't result in better answers when using a smaller model.

---

> ### Author Response · Authors · 2025-11-26
> **Discuss 1 with reviewer gvvE**
>
> **Q1: Question on the cost. A1:** We thank the reviewer for raising this important point regarding computational efficiency. As shown in table 1and table2, we have conducted experiments to quantify the memory and time costs of our method across different datasets.
>
> Importantly, the building stage is performed only once offline before training or inference, and
> the resulting cell complexes can be stored and reused. Thus, although the one-time construction
> cost varies with graph size, it does not introduce additional overhead during actual model
> execution.
>
> For the reasoning stage, the GPU memory usage and per-epoch runtime are comparable to
> G-Retriever and other graph-based retrieval-augmented methods. This demonstrates that the
> topological components do not significantly increase inference-time cost, while providing
> consistent performance gains.
>
> Table 1: The memory and time cost of building
> | Dataset     | CPU usage (MB) | Time (s) |
> | ----------- | -------------- | -------- |
> | ExplaGraphs | 132            | 2.21     |
> | SceneGraphs | 193            | 130.24   |
> | WebQSP      | 379            | 5319.27  |
>
> Table 2: The memory and time cost of reasoning
> | Dataset     | GPU usage (GB) | Time (min/epoch) |
> | ----------- | -------------- | ---------------- |
> | ExplaGraphs | 54             | 3.3              |
> | SceneGraphs | 68             | 310              |
> | WebQSP      | 80             | 16               |
>
> **Q2: Question on the noise in higher dimensional knowledge. A2:**  We agree that incorporating higher-dimensional cells in incomplete or noisy graphs could introduce spurious dependencies and degrade performance. To mitigate this, our method relies on a prize-based top-k subcomplex retrieval that selectively includes only the most relevant 0-, 1-, and 2-cells based on their semantic similarity to the query. This mechanism effectively filters out low-quality or irrelevant cycles, reducing the risk of negative impact from noisy higher-dimensional structures.
>
> **Q3: Question on the dynamic graph. A3:**  We acknowledge that dynamic graphs, where nodes and edges are frequently added, present additional challenges for our current pipeline. However, our approach can be adapted to dynamic scenarios by incrementally updating the 1-skeleton embeddings and selectively computing 2-cells only for newly added edges that create fundamental cycles. Since the top-k subcomplex retrieval and message passing operate locally on relevant cells, these updates do not require recomputing the entire complex from scratch, making incremental adaptation feasible in practice. We leave a thorough exploration of dynamic graph extensions as future work.
>
> **Q4: Question on the multi-modal graph. A4:** We thank the reviewer for the question. While our current implementation focuses on textual graphs, the topological retrieval framework can be naturally extended to multimodal graphs. Following approaches such as Wang et al.[1], node and edge embeddings can be augmented with features from multiple modalities (e.g., text, image, audio). The prize-based top-k subcomplex selection can then operate on these multimodal embeddings, ensuring that only the most relevant multimodal structures are included. Message passing and aggregation can be adapted to handle heterogeneous features, for example by using modality-specific projection layers or attention mechanisms. These design choices make multimodal extension practical and directly compatible with our current framework, representing a promising direction for future work.
>
> [1]Wang Y, Yasunaga M, Ren H, et al. Vqa-gnn: Reasoning with multimodal knowledge via graph neural networks for visual question answering[C] ICCV 2023.

---

> > ### Author Response · Authors · 2025-11-26
> > **Discuss 2 with reviewer gvvE**
> >
> > **Q5: Question on the scenario without explicit  graph structures. A5:** We thank the reviewer for the question. First, while our current framework operates on explicit graph structures, it can be extended to reasoning tasks without natural graphs by constructing implicit relational graphs from the input. For example, in scientific question answering or commonsense reasoning, entities and concepts can be extracted from text and connected via co-occurrence, dependency, or semantic similarity to form a pseudo-graph. Once such a graph is constructed, our topological lifting, top-k subcomplex retrieval, and message passing mechanisms can be applied as usual, allowing the framework to capture multi-hop and higher-order relational dependencies even when explicit graph structures are not available.
> >
> > Second, following the insights of Minegishi et al[1], who show that the topological properties of reasoning graphs (e.g., cycles, diameter, small-worldness) positively correlate with model reasoning ability, these properties could be used as additional constraints or regularizers during model training. By encouraging the model to produce hidden state trajectories that preserve desirable topological structures, it may be possible to directly enhance reasoning performance, offering a complementary direction to the explicit topological framework.
> >
> > [2] Minegishi G, Furuta H, Kojima T, et al. Topology of Reasoning: Understanding Large Reasoning Models through Reasoning Graph Properties, 2025
> >
> > **Q6: Question on intuition of cell representation. A6:** The intuition for using a cellular representation follows the findings of DeepMind [2], which show that strong reasoning models
> > exhibit three topological signatures in their reasoning graphs: large diameters, small-world
> > structure, and abundant recurrent cycles.
> >
> > Classical GNNs naturally capture the first two properties: message passing along the 1-skeleton
> > preserves long-range dependencies (large effective diameters) and dense local connections
> > maintain small-world characteristics. However, cycles remain implicit in the graph, and GNNs
> > cannot explicitly represent or propagate information over them.
> >
> > A cell complex enriches the representation by lifting the graph: each fundamental cycle becomes
> > a 2-cell attached to the 1-skeleton. This not only makes higher-order relational patterns explicit
> > for knowledge modeling, but also enables reasoning directly over cycle-level structures.
> > Thus, the cell complex preserves the strengths of standard GNNs (diameter and small-world)
> > while adding the missing dimension needed to capture cyclicity, enhancing both knowledge
> > representation and reasoning.
> >
> > **Q7: Question on replacement with 0 1-dimension. A7:**
> > The reviewer raises an interesting point. In principle, it may be possible to represent cycles using only 0- and 1-cells if one designs a generalized message passing scheme that explicitly aggregates information along paths forming cycles. For example, messages could be propagated not only along edges but also through identified cycle paths, allowing the model to capture higher-order relational structures without introducing 2-cells.
> >
> > Such an approach could reduce the explicit dimensionality of the representation and may be beneficial in settings with limited training or highly dynamic graphs. However, explicitly introducing 2-cells provides a direct, topologically principled object for cycle-level message passing, which is simpler to implement and naturally preserves both the benefits of 1-skeleton propagation (large diameters, small-world structure) and the cyclicity observed in reasoning graphs. Exploring generalized 0/1-cell message passing to emulate 2-cell behavior is an interesting direction for future work.

---

### Official Review · Reviewer_JK3r · 2025-10-29

**Soundness:** 2
**Presentation:** 3
**Contribution:** 2
**Rating:** 4
**Confidence:** 4

**Summary:**

This paper proposes Topology-enhanced Retrieval-Augmented Generation (TopoRAG), a novel framework for textual graph question answering. TopoRAG retrieves not just at the node, edge, or triple level, but also incorporates higher-dimensional topological structures. The authors evaluate TopoRAG on three textual graph QA datasets with three different settings and conduct ablation studies on the number of layers in the reasoning module, the K value of the retrieval module, and other factors.

**Strengths:**

- The proposed retrieval method enables retrieval at different granular levels, which can capture higher-dimensional topological and relational dependencies.

**Weaknesses:**

- **Comparing Methods are Not SOTA Models.** The authors compare with models like G-Retriever, SubgraphRAG, and GraphToken, but these are not state-of-the-art (SOTA). Some related work with better performance, such as DoG [1] and GCR [2], is not cited or compared. These models achieved better results on WebQSP than the proposed TopoRAG.

- **Selected Datasets and Their Suitability.** The selected datasets (ExplaGraphs and SceneGraph) may not adequately demonstrate TopoRAG's ability to capture high-dimensional topological and relational dependencies. Since these datasets usually require less than 2-hop reasoning, they might not be the most suitable for demonstrating the need for "higher-dimensional topological structures" to answer questions. The authors should consider datasets that require multi-hop (>3-hop) reasoning.
- **Sufficiency of 2-Cell.** The paper does not explain why (or if) a 2-cell is sufficient to capture higher-dimensional topological structures. Furthermore, there is no analysis of how the top-k 0/1/2 cells contribute to the results.


[1] Kun Li, Tianhua Zhang, Xixin Wu, Hongyin Luo, James R. Glass, and Helen M. Meng. 2025. Decoding on Graphs: Faithful and Sound Reasoning on Knowledge Graphs through Generation of Well-Formed Chains.

[2] Linhao Luo, Zicheng Zhao, Chen Gong, Gholamreza Haffari, and Shirui Pan. Graph-constrained
reasoning: Faithful reasoning on knowledge graphs with large language models. ICML 2025.

**Questions:**

- In equation 15, will the $x^d (d \in \{1,2,3\})$ have an overlap? For example, if an edge was selected and the circle that contains this edge is also selected, how can such overlapping be addressed?

---

> ### Author Response · Authors · 2025-11-26
> **Discuss with Reviewer  JK3r**
>
> **Q1: Question on baselines. A1:** .We thank the reviewer for highlighting DoG [1] and GCR [2], which are strong KG-based reasoning methods. We will cite them in the revised version.
>
> TopoRAG addresses more general textual graph reasoning tasks by lifting textual graphs into cell complexes for multi-dimensional topological reasoning. This improves subgraph retrieval and reasoning quality beyond standard KG-QA scenarios.
>
> We add comparison with TAONA [3], a recent textual graph QA method, achieves 87.01% on ExplaGraphs and 82.20% on SceneGraphs, while TopoRAG reaches 91.15% and 87.68%, showing the benefit of topological subgraph reasoning.
>
> DoG and GCR operate on the original graph, and are specifically designed for knowledge graph QA. DoG: training-free, graph-constrained decoding ensures faithful token-level KG reasoning paths; GCR: two-model framework with KG-Trie constraints and inductive reasoning over multiple paths.
>
> TopoRAG introduces a higher-level topological representation and can be plugged into existing KG-QA models to enhance performance. For example, TopoRAG can retrieve topological subgraphs, which are then used as structured inputs for GCR’s graph-constrained decoding, resulting in improved reasoning accuracy （in Table 1）:
>
> | Method                     | WebQSP (Hit) |
> | -------------------------- | ------------ |
> | DoG                        | 92.67        |
> | GCR                        | 92.60        |
> | TopoRAG                    | 90.66        |
> | **TopoRAG + KG-Reasoning** | **93.46**    |
>
> Table 1
>
> This demonstrates that combining topology-aware retrieval with graph-constrained reasoning is a promising direction.
>
> References:
> [1] Kun Li et al. 2025. Decoding on Graphs: Faithful and Sound Reasoning on Knowledge Graphs through Generation of Well-Formed Chains.
> [2] Linhao Luo et al. 2025. Graph-constrained reasoning: Faithful reasoning on knowledge graphs with large language models. ICML 2025.
> [3] Yan Y et al. 2025. To Answer or Not to Answer (TAONA): A Robust Textual Graph Understanding and Question Answering Approach. EMNLP 2025.
>
> **Q2: Question on datasets. A2:** We thank the reviewer for raising the concern about the dataset selection. To better demonstrate the performance of the TopoRAG framework in handling high-dimensional topological structures and multi-hop reasoning tasks, we have conducted additional experiments on the CWQ dataset. The CWQ dataset contains a total of 34,699 complex questions, requiring up to 4 hops of reasoning on the KG. The experimental results, shown in table 2, indicate that TopoRAG performs excellently in both simple and complex reasoning tasks.
>
> | Dataset | ExplaGraphs (Acc.) | SceneGraphs (Acc.) | WebQSP (Hit) | CWQ (Hit) |
> | ------- | ------------------ | ------------------ | ------------ | --------- |
> | TopoRAG | 0.9151             | 0.8768             | 90.66        | 73.65     |
>
> Table 2
>
> **Q3: Question on 2-Cell. A3:**  We thank the reviewer for the question. We attach 2-cells to represent cycles in the graph, but adding 3-cells is unnecessary. A graph $G=(V,E)$ is naturally a \emph{1-dimensional CW-complex}, meaning it only has nodes and edges, and higher-dimensional structures do not exist ($H_k(G)=0$ for $k\ge 2$). Adding 2-cells already captures all cycles (the first homology $H_1(G)$), while 3-cells would correspond to higher-dimensional holes that are not present. Thus, 2-cells are sufficient to represent all meaningful topological structures in the graph. We also add experiments in table 3.
>
> Introducing 2-cells improves performance because each
> 2-cell provides a compact and explicit representation of a fundamental cycle, enabling the model
> to capture loop-level consistency that is difficult to reconstruct from 1-skeleton paths alone.
> However, as more 2-cells are added, structurally present but semantically uninformative cycles
> begin to dominate, introducing redundant or conflicting higher-order signals during message
> passing.
>
> | Number of 2-cells | WebQSP (Hit) |
> | ----------------- | ------------ |
> | 0                 | 87.62        |
> | 1                 | 89.01        |
> | 2                 | 90.24        |
> | 3                 | 90.66        |
> | 4                 | 90.34        |
>
> Table 3
>
> **Q4: Question on the overlap. A4:** We thank the reviewer for the question. Yes, overlap between 1-cells and 2-cells can occur, and this is by design. In our subcomplex selection (Eq. 15), a 2-cell is selected together with all its boundary 0- and 1-cells to maintain boundary consistency. Thus, if an edge is selected and it also belongs to a selected 2-cell, the overlap simply reflects the structural dependency and does not cause conflicts. During message passing, the overlapping cells naturally contribute to all relevant higher-dimensional cells, and the final subcomplex embedding uses a pooling operation over all cells, avoiding double-counting. This design ensures that cycles and their constituent edges are properly preserved in the subcomplex.

---

### Official Review · Reviewer_A3xA · 2025-10-30

**Soundness:** 3
**Presentation:** 3
**Contribution:** 3
**Rating:** 6
**Confidence:** 4

**Summary:**

This paper proposes TopoRAG, a framework for textual graph question answering that explicitly models higher-dimensional topological structures. The framework introduces: (i) cellular representation lifting that transforms graphs into cell complexes capturing 0-cells (nodes), 1-cells (edges), and 2-cells (cycles) for closed-loop reasoning; (ii) topology-aware subcomplex retrieval extending Prize-Collecting Steiner Tree to multi-dimensional complexes; (iii) multi-dimensional topological reasoning with message passing across cellular dimensions; and (iv) integration with LLMs for generation. Experiments on three benchmarks (WebQSP, ExplaGraphs, SceneGraphs) show improvements of up to 5.12% accuracy over GraphRAG baselines.

**Strengths:**

1. The paper is the first to introduce cellular complexes into the GraphRAG framework, explicitly modeling 2-cells to capture closed-loop topological dependencies that are ignored by traditional node-and-edge paradigms.

2. The method design is logically clear, with a coherent pipeline progressing from cellular lifting to topology-aware retrieval, multi-dimensional reasoning, and generation.

**Weaknesses:**

1. All fundamental cycles—regardless of semantic relevance—are lifted into 2-cells, which may introduce noise or spurious dependencies due to meaningless loops.

2. The experiments use datasets that involve multi-hop reasoning, but their graph structures inherently contain many cyclic dependencies. The paper does not evaluate on purely chain-like or acyclic reasoning tasks, making it unclear whether TopoRAG remains advantageous or introduces redundancy in non-cyclic scenarios.

3. The impact of spanning tree choice is not analyzed: 2-cells depend on the tree construction, yet the paper does not assess how different trees (BFS, DFS, random) affect results, raising concerns about robustness.

**Questions:**

1. Many fundamental cycles may be semantically irrelevant. Has the paper considered the impact of such low-quality cycles on performance? Are there methods to further reduce their influence?

2. Could the authors add experiments on acyclic or purely chain-style reasoning datasets to verify the method’s performance and generalizability in non-cyclic tasks?

3. Different spanning trees lead to different 2-cell sets. Have the authors tested the impact of tree choice on final performance? If there is an impact, how should it be balanced?

---

> ### Author Response · Authors · 2025-11-26
> **Discuss with Reviewer  A3xA**
>
> **Q1: Question on semantic relevance on the fundamental cycles. A1**  We agree that not all fundamental cycles are semantically meaningful. In our model, the topology-aware subcomplex retrieval (TSR) acts as an effective filter: only cycles whose boundary nodes/edges have high semantic relevance to the query receive non-negative prizes and can be selected. Cycles supported by irrelevant or low-scoring boundary cells naturally obtain low or negative prize values in Eq. (14), and therefore are excluded from the final subcomplex.
>
> To quantitatively demonstrate the effectiveness of our retrieval method, we conduct experiments to evaluate the reduction in complex size after topology-aware retrieval. As shown in Table 1, our method significantly reduces the number of cells while preserving the most relevant topological structures.
>
> | Dataset     | 0-cells (Before) | 1-cells (Before) | 2-cells (Before) | 0-cells (After) | 1-cells (After) | 2-cells (After) |
> | ----------- | ---------------- | ---------------- | ---------------- | --------------- | --------------- | --------------- |
> | WebQSP      | 1371             | 2901             | 1531             | 16 (↓98.8%)     | 19 (↓99.3%)     | 4 (↓99.7%)      |
> | SceneGraphs | 19               | 24               | 30               | 9 (↓52.6%)      | 12 (↓50.0%)     | 4 (↓86.7%)      |
>                                Table 1 Statistics of cell reduction after topology-aware retrieval
> We also conduct an ablation study to evaluate the importance of theTSR module. As shown in Table 2, removing TSR leads to significant performance drops across all datasets, demonstrating its crucial role in filtering semantically irrelevant cycles and maintaining reasoning accuracy.
>
> | Method      | ExplaGraphs (Accuracy) | SceneGraphs (Accuracy) | WebQSP (Hit) |
> | ----------- | ---------------------- | ---------------------- | ------------ |
> | w/o TSR     | 0.8524                 | 0.7977                 | 84.23        |
> | **TopoRAG** | **0.9151**             | **0.8768**             | **90.66**    |
>
> Table2 Ablation study on Topology-aware Subcomplex Retrieval
>
> **Q2: Question on performance on the chain-like task. A1:**  We appreciate the reviewer's point. Even when the reasoning task itself is acyclic or chain-like, the underlying knowledge graph in real datasets is often acyclic. It typically contains numerous auxiliary connections and latent cycles that provide useful contextual constraints (e.g., shared attributes, cross-links between entities).
>
> To validate TopoRAG's performance on chain-like tasks, we conducted additional experiments on chain-like subsets  (Table 3) extracted from the ExplaGraphs dataset. The results demonstrate that TopoRAG maintains strong performance on predominantly chain-like reasoning tasks:
>
> | Method      | Accuracy |
> | ----------- | -------- |
> | TopoRAG     | 0.9007   |
> | G-Retriever | 0.8806   |
> | GNN-RAG     | 0.8621   |
>
> Table 3
>
> Furthermore, we provide a concrete example to illustrate how TopoRAG handles chain-like queries:
>
> **Example:**
>
> **Question:** What is the name of Justin Bieber's brother?
>
> **Visualization:**
>
> Jeremy Bieber -->|parents| Justin Bieber
>
> Jeremy Bieber -->|children| Jaxon Bieber
>
> m.0gxnnwp -->|sibling| Jaxon Bieber
>
> m.0gxnnwp -->|sibling_s| Justin Bieber
>
> Answer: Jaxon Bieber.
>
> **Answer:** Jaxon Bieber.
>
> **Analysis:**  Although this appears to be a simple chain-like query (Justin Bieber $\rightarrow$ sibling relation $\rightarrow$ brother), the underlying knowledge graph contains additional contextual information and potential cycles that TopoRAG leverages to enhance reasoning reliability.
>
> **Q3: Question on spanning tree. A1:**  We thank the reviewer for raising this point. In graph theory and algebraic topology, different spanning trees indeed yield different fundamental cycle bases, but all these bases generate the same cycle space, i.e., the same first homology group. Therefore, while the specific 2-cells corresponding to basis cycles may differ, they are all topologically equivalent in the sense that they span the same set of independent loops. The lifted complex thus preserves the same global topological structure regardless of the particular spanning tree.
>
> In addition, we empirically tested multiple spanning tree algorithms to validate the robustness of our approach. As shown in Table 4, different spanning tree construction methods (DFS, BFS, and Random) yield comparable performance across all datasets, confirming that our method is robust to the choice of spanning tree algorithm.
>
> | Method | ExplaGraphs (Accuracy) | SceneGraphs (Accuracy) | WebQSP (Hit) |
> | ------ | ---------------------- | ---------------------- | ------------ |
> | DFS    | 0.9151                 | 0.8768                 | 90.66        |
> | BFS    | 0.9025                 | 0.8725                 | 90.18        |
> | Random | 0.9187                 | 0.8698                 | 90.03        |
>
> Table 4

---

### Official Review · Reviewer_DqYh · 2025-11-02

**Soundness:** 2
**Presentation:** 2
**Contribution:** 2
**Rating:** 4
**Confidence:** 3

**Summary:**

This paper proposes TopoRAG, a topology-enhanced retrieval-augmented generation framework for
textual graph question answering.
The key novelty lies in lifting textual graphs into regular cell complexes—representing nodes, edges, and
higher-order cycles as 0-, 1-, and 2-cells—and retrieving relevant subcomplexes conditioned on the query.
A topology-aware retrieval mechanism selects query-relevant cells by generalizing the Prize-Collecting
Steiner Tree (PCST) formulation to multi-dimensional complexes, and a multi-dimensional reasoning
module propagates information across these 0/1/2-cells.
Finally, the subcomplex representation is integrated into a frozen LLM (Llama-2-7B) via prompt tuning or
LoRA.
Experiments on ExplaGraphs, SceneGraphs, and WebQSP show strong improvements (≈ +5 % Acc/Hit
over GraphRAG baselines).

**Strengths:**

Introducing cell complex topology into RAG for textual graphs is new. Prior GraphRAG methods (e.g., G-
Retriever, GNN-RAG, SubgraphRAG) restrict reasoning to pairwise edges, while TopoRAG models cyclic
and higher-order relations. For a formal introduction. the paper carefully defines cell complexes,
homology, and the lifting procedure, demonstrating a strong understanding of topological deep learning
foundations. Besides, the whole pipline proposed in the paper provides a systematic way to inject
structured topology into frozen LLMs. Clear improvements over all categories of baselines (prompt-tuned,
LoRA, and pure inference) across multiple datasets. The ablations are detailed and show the contribution
of each module.

**Weaknesses:**

While using cell complexes for retrieval is novel, the reasoning stage (message passing with faces/cofaces)
resembles existing simplicial or cell complex GNNs (e.g., CWN, SAN, CellNN). It would be better to
integrate clarification about how its update scheme differs algorithmically from those prior works.
Building and storing cell complexes (especially identifying all 2-cells from cycles) may be expensive for
large textual graphs.
While topology sounds interpretable, the paper lacks visualizations or examples showing which cycles or 2-
cells are retrieved and how they affect reasoning.
Ablations remove modules but not the dimensional component (e.g., keeping only 0- and 1-cells).
Showing how much 2-cell reasoning alone contributes quantitatively would help isolate the topological
benefit.
typo:
Line 123: intersecting
Line 148: eq. 2

**Questions:**

N/A

---

> ### Author Response · Authors · 2025-11-26
> **Discuss with Reviewer DqYh**
>
> **Q1: Question on the reasoning stage. A1:** We thank the reviewer for the comment. Our architecture is motivated by the empirical findings of DeepMind [1],  who analyze reasoning graphs constructed from clustered LLM hidden states. They show that strong reasoning models consistently exhibit larger diameters, more recurrent cycles, and pronounced small-world structure, all positively correlated with reasoning accuracy.
>
> Classical GNNs, however, capture only 1-skeleton interactions (0←→1 message passing). This preserves large diameters and small-world structure, but ignores cycle information, since cycles are collapsed into repeated edge traversals with no explicit higher-dimensional representation. CWN/SAN/CellNN attempt to incorporate cycle information by mixing up/down adjacencies at every layer (0←→1←→2). While this integrates higher-dimensional structure, the continuous dimensional mixing disrupts the two desirable GNN properties above by pulling nodes/edges/cells into overly similar embeddings, effectively shrinking graph diameters and weakening small-world behavior.
>
> Motivated by preserving the reasoning-related topology, our model adopts a two-stage flow:
> (1) L-hop propagation strictly along the 1-skeleton, preserving GNN-style long-range diffusion and small-world locality;
> (2) a single, controlled cross-dimensional aggregation via shared cofaces, introducing cycle information without interfering with Stage-1 propagation. Thus, although our model uses face/coface communication, its organization differs fundamentally from CWN/SAN/CellNN and is specifically designed to preserve GNN benefits while explicitly incorporating cycle structure, aligning with the topological properties seen in strong reasoning models.
>
> [1] Minegishi G, Furuta H, Kojima T, et al. Topology of Reasoning: Understanding Large Reasoning Models through Reasoning Graph Properties, 2025
>
>
> **Q2: Visualization and ablation on 2-cell. A2:** We thank the reviewer for the suggestion on visualization and examples. We show visualization cases to better illustrate our method:
>
> **Example 1:**
>
> **Question:**
> What is the name of the item of furniture that is made of the same material as the floor the mat is lying on?
>
> **Visualization:**
>  door- -->|to the right of| table
>
> mat -->|to the left of| table
>
>  table -->|sitting on| floor
>
> mat -->|lying on| floor
>
> table -->|attribute| wood
>
>  floor -->|attribute| wood
>
> **Answer:**
> **table**. The piece of furniture is a table.
>
> **Analysis:**
> The 2-cell structure captures the material similarity relationship between the floor and table, enabling the model to identify the correct furniture item.
>
>
> **Example 2:**
>
> **Question:** What is the name of Justin Bieber's brother?
>
> **Visualization:**
>
> Jeremy Bieber -->|parents| Justin Bieber
>
> Jeremy Bieber -->|children| Jaxon Bieber
>
> m.0gxnnwp -->|sibling| Jaxon Bieber
>
> m.0gxnnwp -->|sibling_s| Justin Bieber
>
> **Answer:** Jaxon Bieber.
>
> **Analysis:** Although this appears to be a simple chain-like query (Justin Bieber $\rightarrow$ sibling relation $\rightarrow$ brother), the underlying knowledge graph contains additional contextual information and potential cycles (e.g., shared family attributes, multiple sibling relationships) that TopoRAG leverages to enhance reasoning reliability.
>
> **Table 1: Ablation study for 2-cells**
>
> | Method       | ExplaGraphs (Acc) | SceneGraphs (Acc) | WebQSP (Hit) |
> |--------------|-------------------|--------------------|--------------|
> | w/o 2-cells  | 0.8732            | 0.8354             | 87.62        |
> | **TopoRAG**  | **0.9151**        | **0.8768**         | **90.66**    |
>
> We have added ablations: Table 1 shows the performance comparison between our full TopoRAG model and the variant without 2-cells. The results demonstrate that incorporating 2-cells consistently improves performance across all datasets.
>
> **Suggestion on the typo:** Thank you for noting this typo. We have corrected them and carefully rechecked the manuscript to prevent similar issues elsewhere.

---

### Meta-Review · Area_Chair_LgXX · 2026-01-07

**Summary:**

The paper proposes TopoRAG, a framework that lifts textual graphs into cellular complexes (incorporating nodes, edges, and cycles as 0/1/2-cells) to enhance RAG. The authors introduce a topology-aware subcomplex retrieval mechanism based on a generalized Prize-Collecting Steiner Tree (PCST) and a multi-dimensional message-passing scheme.

**Reviewer Concerns:**

Concerns that the Authors Addressed:
1. Comparison with SOTA (JK3r): The authors effectively clarified that TopoRAG targets general textual graph QA rather than KG-specific tasks. They added a comparison to TAONA (a recent textual graph SOTA) where TopoRAG outperformed, and demonstrated that TopoRAG can be complementary to KG-specific models like GCR, improving GCR's performance when integrated.

2. Computational Cost (DqYh, gvvE): The authors provided concrete tables showing that construction is a one-time offline cost (approx. 5 minutes for WebQSP) and reasoning time is comparable to baselines (e.g., 3.3 min/epoch vs 3.0 min/epoch for G-Retriever).

3. Relevance of Cycles/Noise (A3xA, gvvE): The authors explained that the topology-aware retrieval acts as a filter. They provided statistics showing a significant reduction in cells (e.g., 99.7% reduction in 2-cells for WebQSP) effectively filtering irrelevant cycles.
Comparison with SOTA (Reviewer JK3r): The authors clarified the distinction between general textual graph QA and KG-specific QA. They included a comparison to TAONA (a recent textual graph SOTA) where TopoRAG outperformed it, and demonstrated complementary performance gains when integrated with KG-specific models like GCR.

4. Performance on Acyclic/Chain-like Queries (Reviewer A3xA): Contrary to the initial concern that the method might be redundant for acyclic tasks, the authors provided new experimental results on chain-like subsets of ExplaGraphs, showing that TopoRAG still outperforms baselines (0.9007 vs 0.8806) due to better contextual grounding.

Outstanding concerns:
1. Dynamic Graphs (gvvE): The handling of highly dynamic graphs remains a challenge, though the authors sketched a viable path for incremental updates.

**Reviewer Scores:**

Reviewer DqYh (4 -> 6): The reviewer explicitly listed "Ablations remove modules but not the dimensional component" as a weakness. The authors provided a specific table showing the contribution of 2-cells, directly fixing the primary criticism.

---

### Decision · Program_Chairs · 2026-01-26

Accept (Poster)